# O-Mem: Omni Memory System for Personalized, Long Horizon, Self-Evolving Agents

## Abstract

Recent advancements in LLM-powered agents have demonstrated significant potential in generating human-like responses; however, they continue to face challenges in maintaining long-term interactions within complex environments, primarily due to limitations in contextual consistency and dynamic personalization. Existing memory systems often depend on semantic grouping and the retrieval of past interaction groupings, which can overlook semantically irrelevant yet critical user information and introduce retrieval noise. To address these issues, we propose O-Mem, a novel memory framework based on active user profiling that dynamically extracts and updates user characteristics and event records from interactions. O-Mem supports hierarchical retrieval of persona attributes and topic-related context, enabling more adaptive and coherent personalized responses. Additionally, we introduce a new dataset designed to evaluate personalized long-text generation in memory-augmented agents. Experiments across three personalized tasks demonstrate that O-Mem consistently improves long-term human–AI interaction by scaling memory-time within interactions.

## 1 Introduction

LLM-empowered agents have demonstrated huge potential in generating human-level intelligent responses (Schlegel et al., 2025) but still lacks long-term interaction ability with complex external environments (Zhang et al., 2024). This limitation causes agents struggle to maintain consistency of context across time (Laban et al., 2025) and reduced their personalization capability dynamically adapting to users' situations (Zhang et al., 2025).

Motivated by practical demands, multi agent memory systems have been proposed in recent years. These systems store user past interactions in diverse architectures and enable agents to retrieve relevant information from them to deliver more personalized responses. For instance, Memory OS (Kang et al., 2025) categorizes user interactions into short-term, mid-term, and long-term persona memory caches based on timestamps and frequency of occurrence. Agentic Memory (Xu et al., 2025) organizes interactions into distinct boxes according to their semantic similarity, while Mem0 (Chhikara et al., 2025) extracts meaningful content from messages and stores it in a database to support semantic retrieval based on user queries. By structuring user information more effectively, these systems enhance the ability of agents to provide efficient and highly personalized responses.

The core pipeline of such memory systems involves categorizing received messages into groups based on topics and retrieving relevant memory groupings during user interactions. However, this design presents several significant shortcomings: i) Memory systems that rely heavily on semantic retrieval may overlook information that is semantically irrelevant yet potentially important—such as broader user characteristics or situational context—which is crucial for interactions requiring a comprehensive understanding of the user. As illustrated on the left side of Figure 1, an intelligent agent should consider the user's health condition and recent schedule when planning weekend activities, rather than relying solely on activity-related memories. ii) The chunk-based retrieval architecture may introduce additional retrieval noise. As shown on the right side of Figure 1, unintended memory grouping may compel the agent to retrieve information from all three groupings to gather sufficient context for appropriate user interaction. These redundant retrievals diminish the effectiveness of the model's responses, while also increasing interaction latency and token consumption during large language model inference.

Figure 1: Left: Comparison between the conventional memory system with semantic retrieval from message groupings (in purple) and our proposed user-centric framework O-Mem employing characteristic identification, event recording, and topic-message indexing (in blue). Right: Comparison between expected user message clustering and actual inappropriate grouping results from conventional Chunk-Retrieve memory systems.

In this paper, we propose O-Mem, a memory framework based on active user profiling. Unlike conventional approaches that merely store and group past interactions for retrieval, O-Mem actively extracts and updates user persona characteristics and event records from ongoing dialogues. This enables the agent to progressively refine its understanding of the user's attributes and historical experiences. We redefine the retrieval process in personalized memory systems by treating each interaction as a query for relevant persona attributes and past events—effectively leveraging both persona profiles and topic-related interactions as contextual cues to support personalized responses. To facilitate research in personalized memory systems, we introduce a newly constructed and annotated dataset,Personalized Deep Research Bench, specifically designed for personalized long-text generation tasks. Unlike existing datasets, Personalized Deep Research Bench is composed of real-world dialogues between human users and agents.

- We identify limitations in existing chunking-based semantic retrieval memory frameworks, notably their inadequate user understanding and restricted personalization abilities. To overcome these issues, we explore an automated mechanism that constructs a memory system offering more comprehensive and adaptive personalization.

- We propose O-Mem, a novel persona memory framework that utilizes dynamic user profiling and a hierarchical, user-centric retrieval strategy. Unlike approaches that rely solely on semantic retrieval of past messages, O-Mem actively constructs and updates user profiles by accumulating knowledge from interaction histories. Furthermore, we introduce Personalized Deep Research Bench, the first dataset specifically designed for evaluating memory capabilities in personalized long-text generation.

- Extensive experiments on three persona-oriented tasks—i) persona-based open question answering, ii) persona-guided response selection, and iii) persona-centric in-depth report generation—show that O-Mem consistently improves performance across a variety of personalized applications. By enabling dynamic user profiling and **memory-time scaling**, O-Mem allows LLM agents to continuously adapt to users' evolving needs, demonstrating strong potential for enhancing long-term human-AI interactions through more coherent personalized responses.

## 2 LITERATURE REVIEW

**Agent Memory Sysytem** Intelligent agent systems powered by large language models (LLMs) have gained significant popularity in recent years owing to their remarkable capabilities in task comprehension and execution (Shen & Yang, 2025; Fırat & Kuleli, 2023; Wu et al., 2024). Nevertheless, these systems continue to grapple with the challenge of sustaining high-quality performance when incorporating historical experience across complex, long-duration scenarios (Kang et al., 2025; Zhang et al., 2024). To address this limitation, numerous agent memory enhancement frameworks have been proposed, which can be broadly classified into two categories: (i) approaches that fine-tune LLM parameters to enhance information memorization and utilization (Wei et al., 2025; Zhou

et al., 2025; Modarressi et al., 2024), and (ii) methods that employ sophisticated information organization and retrieval techniques within external memory systems to preserve LLMs' long-term capabilities. The latter approach has attracted considerable attention due to its plug-and-play nature, which eliminates the need for additional training costs. Furthermore, these methods significantly reduce the dependency of memory capacity on the LLMs' input window length. For example, Think-in-Memory (TiM) (Liu et al., 2023) preserves the reasoning traces of LLMs across multiple dialogue rounds to alleviate response inconsistencies. A-mem (Xu et al., 2025) organizes memory fragments into linked lists to improve retrieval performance. Grounded Memory (Ocker et al., 2025) introduces vision language models (VLMs) to interpret consecutive audio frames, organizing these interpretations in a graph structure for subsequent retrieval. MemoryBank (Zhong et al., 2024) incorporates the Ebbinghaus Forgetting Curve theory to enable agents to forget and reinforce memories based on time elapsed and the relative significance of memory segments. MemGPT (Packer et al., 2023) and Memory OS (Kang et al., 2025) adopt an operating system-like architecture for memory organization and retrieval, employing mechanisms such as a first-in-first-out queue for working memory management. However, most existing systems overlook a critical aspect: how to dynamically and hierarchically establish connections between memory fragments to continuously update the agent's overall understanding of its environment. For instance, while A-mem (Xu et al., 2025) and Memory OS (Kang et al., 2025) store semantically similar information in linked segments and retrieve the grouped data during response generation, their simple chunk-retrieval mechanisms, as illustrated in Figure 1, often fail to equip agents with a comprehensive and in-depth understanding of users prior to interaction. Therefore, to bridge this gap, we propose a novel memory system, O-Mem, based on active user profiling. The key difference between O-Mem and previous memory systems is that its core task is to answer the questions: "What kind of person is this user? What has he or she experienced?" rather than merely grouping received user information for later retrieval. We draw inspiration from the human brain's memory architecture and consequently redefine the three core components of an agent's memory: i).**Episodic Memory**, which is responsible for mapping a user's historical interaction cues to their corresponding situational contexts (e.g., mapping the cue "project deadline" to the specific episode where the user expressed stress and requested help with scheduling); ii).**Persona Memory**, which constructs and maintains a holistic profile of the user; iii).**Working Memory**, which is responsible for providing relevant contextual information to the current interaction. Together, these components work in concert to enable O-Mem to build a deep, dynamic understanding of the user, powering more personalized and context-aware interactions.

**Persona Agent.** While large language models (LLMs) serve as powerful assistants for a multitude of tasks, their effectiveness remains constrained without the ability to learn from and adapt to human preferences through personalization. A promising direction involves the development of persona agents—LLM-based systems deeply integrated with personal data to deliver responses aligned with user-specific needs (Li et al., 2024b). Meeting the growing demand for such personalized interactions requires methodologies that can continuously and accurately infer user characteristics from their interactions (Sun et al., 2024; Magister et al., 2024; Eapen & Adhithyan, 2023). Most prior work on persona-enhanced LLMs has focused on injecting user information through fine-tuning (Salemi et al., 2024; Eapen & Adhithyan, 2023; Tan et al., 2024) or direct retrieval from user traces of static user profiles that rely on a limited set of predefined attributes(Richardson et al., 2023; Sun et al., 2024; Qiu et al., 2025). However, these approaches face significant limitations in handling long-term, dynamic and evolving user preferences: fine-tuning requires computationally expensive retraining for each update, while direct retrieval lacks the capacity to synthesize longitudinal interaction patterns into coherent and evolving user profiles. In this work, we propose a persona memory system that dynamically organizes a user's interaction history into structured persona characteristics and experiential data, enabling more precise, adaptive, and personalized responses over time.

## 3 METHOD

To build a robust personalized memory system, it is essential to capture a holistic view of the user's historical events and traits (persona memory), along with contextual information from past interactions (working memory). The system should also support recall of specific user experiences based on situational clues (episodic memory). As illustrated in Figure 2, O-Mem implements these three memory modules via specialized components and integrated retrieval mechanisms. It continuously extracts and refines user profiles through ongoing interaction, building a semantic mapping between topics/clues and corresponding interaction scenarios. This architecture enables dynamic, multi-

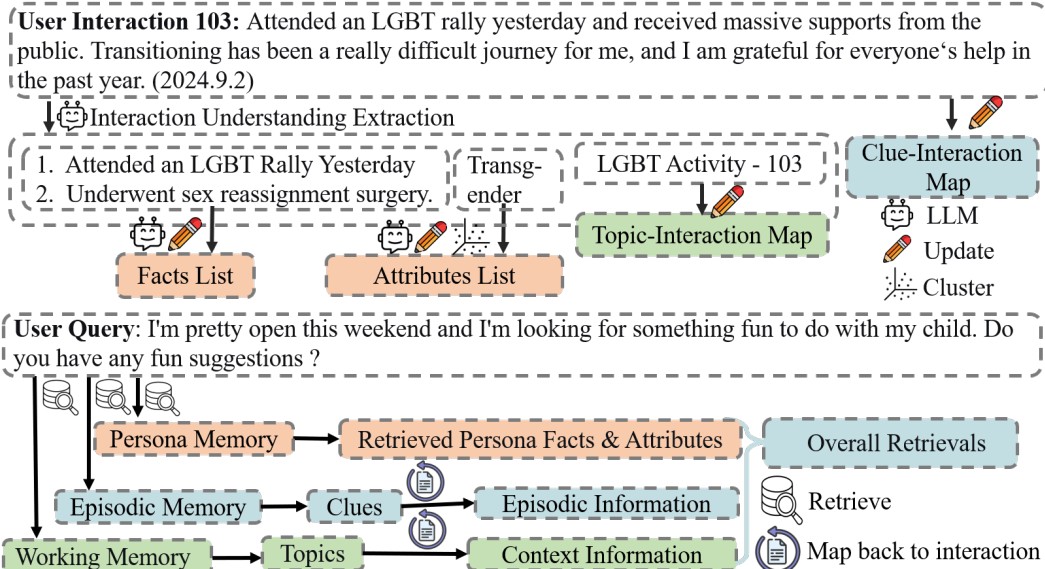

Figure 2: Top: The process of encoding user interactions into memory in O-Mem. Different colors refers to different memory components. O-Mem encodes a user interaction into memory by extracting and recording relevant user attributes and event data into persona memory, episodic memory, and working memory. Bottom: The memory retrieval process concerning one user interaction in O-Mem. O-Mem retrieves from all its three memory components concerning one new user query.

faceted user understanding that supports powerful personalization. In this section, we first present the notation used in our work (subsection 3.1), followed by an explanation of how user interactions are encoded into the different memory components in O-Mem (subsection 3.2), and finally describe the retrieval process across these memories (subsection 3.3).

## 3.1 PRELIMINARY: NOTATION

In this section, we define a user interaction, denoted as $U$, as a record of either explicit literal content (e.g., search queries) or implicit user behavior (e.g., taking a screenshot). Let $M_w$ be a dictionary that maps each clue word $w$ to the interactions in which it appears, and $M_t$ be a dictionary that maps each topic to its corresponding interactions. Additionally, $P_a$ denotes the list of persona attributes, and $P_f$ represents the list of persona fact events.

As illustrated in Figure 2, we model these components within a cognitive architecture: $P_f$ and $P_a$ constitute the **user persona memory**, which stores long-term, abstracted user knowledge; $M_t$ functions as **working memory**, capturing the topical context of the current interaction; and $M_w$ serves as **episodic memory**, acting as an associative index that links salient clues to their originating interactions. Unlike the strict physiological definitions of working memory and episodic memory, **The definitions of agent working memory and episodic memory in O-Mem are past interactions related to the current interaction topic and past interactions related to clues in the current interaction, respectively.** The semantic similarity function $s(t_1, t_2)$ between two text segments $t_1$ and $t_2$, and the memory retrieval function $F_{\text{Retrieval}}$ based on $s$, are formally defined as follows:

$$s(\mathbf{t_1}, \mathbf{t_2}) = \frac{\mathbf{f_e}(\mathbf{t_1}) \cdot \mathbf{f_e}(\mathbf{t_2})}{\|\mathbf{f_e}(\mathbf{t_1})\|\|\mathbf{f_e}(\mathbf{t_2})\|}, \quad F_{\text{Retrieval}}(M \mid q) = \text{top-}k\big\{s(m, q) \mid m \in M\big\}$$

where $f_e(\cdot)$ denotes a text embedding function, top-$k$ returns the $k$ most similar items, and $M$ refers to the memory component from which retrieval is performed.

## 3.2 MEMORY CONSTRUCTION PROCESS

Given the $i$-th user interaction $u_i$, O-Mem first extracts its topic $t_i$, revealed user attribute $a_i$ and past event $e_i$ with large language model $\mathcal{L}$:

$$(t_i, a_i, e_i) = \mathcal{L}(u_i) \tag{1}$$

The clue interaction map $M_w$ and topic interaction map $M_t$ are updated by increasing the count for each word in $u_i$ and $t_i$ :

$$M_t^{(i+1)}[t_i] \leftarrow M_t^{(i)}[t_i] \cup \{i\}, \quad M_w^{(i+1)}[w_j] \leftarrow M_w^{(i)}[w_j] \cup \{i\}, \quad \forall w_j \in \mathcal{T}(u_{(i)}) \tag{2}$$

where $\mathcal{T}(u_{(i)}) = \{w_1, w_2, \ldots, w_n\}$ represents the tokenized words from the $i$-th user interaction. For $e_i$, $\mathcal{L}$ generates an memory management operation regarding its integration with the existing persona fact event list $P_f$:

$$\mathrm{Op}(e_i) \leftarrow \mathcal{L}(e_i, P_f) \in \{\text{Add,Ignore,Update}\}, \quad P_f \leftarrow \mathrm{ApplyOp}(P_f, e_i, \mathrm{Op}(e_i)) \tag{3}$$

where Op refers to the operation decision from $\mathcal{L}$ and ApplyOp refers to the function that executes this operation. During our observation, we identified that similar attributes from the same user frequently recur across different interactions (e.g., users always mention their hobbies repeatedly). To better organize these extracted attributes, we propose an LLM-augmented nearest neighbor clustering method:

$$\mathrm{Op}(a_i) \leftarrow \mathcal{L}(a_i, P_a^t) \in \{\text{Add, Ignore, Update}\}, \quad P_a^t \leftarrow \mathrm{ApplyOp}(P_a^t, a_i, \mathrm{Op}(a_i)) \tag{4}$$

$$\mathrm{NN}(a_i) = \underset{a_l \in P_a^t, l \neq i}{\arg\min} (1 - s(a_i, a_l)) \tag{5}$$

$$G = (V, E), \ V = \{a_1, \ldots, a_K\}, \ E = \{(a_l, \mathrm{NN}(a_l)) \mid a_l \in P_a^t\} \tag{6}$$

$$\mathcal{B} = \{B_1, \ldots, B_M\} = \mathrm{ConnectedComponents}(G), \quad P_a = \bigcup_{m=1}^{M} \mathcal{L}(B_m) \tag{7}$$

where $\mathrm{NN}(a_i)$ denotes the nearest neighbor of attribute $\mathbf{a}_i$ based on the similarity function $s(\cdot, \cdot)$; $G = (V, E)$ represents the nearest-neighbor graph constructed from the temporary attribute list $P_a^t$, with vertices corresponding to attributes and edges connecting each attribute to its nearest neighbor; $K$ refers to the total number of attributes in $P_a^t$, $\mathcal{B} = B_1, \ldots, B_M$ is the set of connected components obtained from $G$ via connected component analysis; and the final attribute set $P_a$ is obtained by applying the large language model $\mathcal{L}$ to analyze the aggregated attributes within each connected component.

## 3.3 MEMORY RETRIEVAL PROCESS

For each user interaction $u_i$, O-Mem conducts retrieval from the user's persona memory, episodic memory, and working memory. We introduce their retrieval process separately.

**Working Memory Retrieval.** we define the retrieval process of working memory as:

$$R_{working} = \bigcup_{t \in \hat{T}} M_t[t], \quad \text{where} \quad \hat{T} = F_{\text{Retrieve}}(\mathcal{K}(M_t), u_i) \tag{8}$$

where $M_t$ is the mapping from topics to their corresponding interactions, $\mathcal{K}(M_t)$ denotes the set of topics in $M_t$, $u_i$ is the current interaction, $F_{Retrieve}$ retrieves the most relevant topics $\hat{T}$ for $u_i$ from $\mathcal{K}(M_t)$, and $R_{working}$ is the set of interactions in $M_t$ corresponding to these relevant topics.

**Episodic Memory Retrieval.** We define the retrieval process of episodic memory as follows. The episodic memory is structured as a word-to-interactions mapping dictionary $M_w$ ($M_w : w \to \{i\}$), which maps words to the sets of past interactions (memory entries) in which they appear. That is, for a word $w$, $M_w[w]$ yields all past interactions containing $w$. Given the current user interaction $u_i$, the retrieval process is: (1) **Tokenization:** Tokenize the utterance into a sequence of words: $W = \mathrm{Tokenize}(u_i)$. (2) **Clue Selection:** Calculate the clue selection score for each word $w \in W$ with respect to the clue-interaction map $M_w$. The word with the highest score is selected as the target clue $\hat{w}$:

$$\hat{w} = \underset{w \in W}{\arg\max} \, \mathrm{Score}(w, M_w) \tag{9}$$

$$\mathrm{Score}(w, M_w) = \frac{1}{df_w} \tag{10}$$

where $df_w$ is the number of past interactions in $M_w$ that contain the word $w$ (i.e., $df_w = |M_w[w]|$). The set of episodic memory interactions associated with the clue $\hat{w}$ is then retrieved as: $R_{episodic} = M_w[\hat{w}]$.

**Persona Memory Retrieval.** We define the persona retrieval process as:

$$R_{persona} = F_{\text{Retrieval}}(P_f, u_i) \oplus F_{\text{Retrieval}}(P_a, u_i) \qquad (11)$$

where $P_f$ refers to the persona facts, $P_a$ refers to the persona attributes, $u_i$ is the current user interaction, $\oplus$ denotes the concatenation operation, and $R_{persona}$ refers to the retrieved persona information.

where $P_f$ refers to the persona facts, $P_a$ refers to the persona attributes, $u_i$ is the current user interaction, $\oplus$ denotes the concatenation operation, and $R_{persona}$ refers to the retrieved persona information.

**Overall Memory Retrieval.** We define the overall retrieval and final response as:

$$R = R_{\text{working}} \oplus R_{\text{episodic}} \oplus R_{\text{persona}}, O = \mathcal{L}(R, u_i) \qquad (12)$$

where $R$ represents the overall retrieved memory content, $O$ represents the final response generated by the language model $\mathcal{L}$ based on the current user interaction $u_i$ and the retrieved memories.

## 4 EXPERIMENT

**Datasets and Evaluation Metrics.** We evaluate our method on three benchmarks: **LoCoMo** (Maharana et al., 2024), **PERSONAMEM** (Jiang et al., 2025), and **Personalized Deep Research Bench** (a new dataset introduced in this work).

The **LoCoMo** benchmark features extended dialogues averaging 300 turns across four memory challenge types: Single-hop, Multi-hop, Temporal, and Open-domain. The **PERSONAMEM** dataset contains user-LLM conversations spanning 15 diverse topics. To address the need for evaluating personalized long-text generation, we introduce **Personalized Deep Research Bench**, a benchmark simulating real-world deep research scenarios. Unlike existing datasets, Personalized Deep Research Bench comprises 50 deep research queries derived from multi-round conversations between 25 real users and LLMs, requiring nuanced understanding of individual user characteristics. It is built upon a subset of a deep research dataset collected from real users by our team, but is specifically repurposed and curated for assessing memory system (see Appendix for details).

For evaluation, we employ: • **LoCoMo**: F1 and BLEU-1 scores following the standard protocol; • **PERSONAMEM**: Accuracy for multiple-choice questions; • **Personalized Deep Research Bench**: Goal Alignment and Content Alignment scores, measuring adherence to user characteristics and expectations via LLM-as-a-judge (Li et al., 2024a). Further details on the evaluation methodology are provided in the appendix.

**Compared Baselines.** Our method is compared with: (i) open-source memory frameworks: A-Mem (Xu et al., 2025), MemoryOS (Kang et al., 2025), Mem0 (Chhikara et al., 2025), and LangMem (Langchain-Ai); and (ii) commercial/proprietary frameworks: ZEP (Rasmussen et al., 2025), Memos (Li et al., 2025), and OpenAI (Mem). **Due to budget constraints and licensing costs**, we report results from original publications for commercial frameworks.

**Implementation Details.** We use all-MiniLM-L6-v2 (all, 2024) as embedding model in O-Mem to calculate similarities. All of our experiments are conducted on two A800 GPUs. The choice of language models across datasets was informed by computational budget. A comparative analysis using both GPT-4.1 and GPT-4o-mini was performed on the LoCoMo benchmark. For the remaining datasets (PERSONAMEM and Personalized Deep Research Bench), only the larger GPT-4.1 model was used.

**Performance Comparison.** The experimental results in three benchmark datasets are separately shown in Table 1, Table 2, and Table 3. Due to limited access to ZEP(Rasmussen et al., 2025), Memos(Li et al., 2025), and OpenAI memory(Mem), we only report their performance reported in their work using GPT-4o-mini. For Personalized Deep Research Bench benchmark dataset, we only compare our method with mem0(Chhikara et al., 2025) and MemoryOS(Kang et al., 2025) for cost efficiency. O-Mem demonstrates superior performance compared to all baselines across three benchmark datasets. The performance advantage is more pronounced in complex reasoning tasks. As shown in Table 1, on the comprehensive LoCoMo benchmark, O-Mem achieves the highest average F1 scores of 51.67% with GPT-4.1 and 50.60% with GPT-4o-mini, outperforming the strongest baselines by significant margins (2.95% and 6.18% absolute improvements, respectively). The performance advantage is particularly pronounced in complex reasoning tasks. For Temporal reasoning, O-Mem achieves F1 scores of 57.48% (GPT-4.1) and 53.54% (GPT-4o-mini), substantially outperforming all baselines. This indicates that our memory management mechanism effectively handles temporal dependencies and sequential information, which is crucial for maintaining coherent long-term conversations. Table 2 further demonstrates O-Mem's effectiveness in personalized interaction scenarios on the PERSONAMEM dataset. O-Mem achieves an average accuracy of 62.99%, exceeding the closest competitor (A-Mem at 59.42%) by 3.57%. Notably, O-Mem excels in challenging tasks such as "Generalize to new scenarios" (73.68%) and "Revisit reasons behind preference updates" (89.90%), highlighting its robust capability in understanding and adapting to evolving user preferences. The superiority of O-Mem is consistently validated on our newly introduced Personalized Deep Research Bench dataset (Table 3), where it achieves an average alignment score of 44.49%, significantly higher than Mem0 (36.43%). This 8.06% improvement demonstrates our method's practical utility in real-world personalized deep research scenarios that require nuanced understanding of individual user characteristics.

Table 1: Performance comparison using different LLMs on Locomo with best scores highlighted.

| LLM | Method | Cat1: Multi-hop | | Cat2: Temporal | | Cat3: Open | | Cat4: Single-hop | | Average | |
|---|---|---|---|---|---|---|---|---|---|---|---|
| | | F1 | B1 | F1 | B1 | F1 | B1 | F1 | B1 | F1 | B1 |
| | LangMemory | 41.11 | 32.09 | 53.67 | 46.22 | **33.38** | **27.26** | 51.13 | 44.22 | 48.72 | 41.36 |
| | Mem0* | 30.45 | 22.15 | 10.69 | 9.21 | 16.75 | 11.34 | 30.32 | 25.82 | 25.40 | 20.78 |
| GPT-4.1 | MemoryOS | 29.25 | 20.79 | 37.73 | 33.17 | 22.70 | 18.65 | 43.85 | 38.72 | 38.58 | 33.03 |
| | A-Mem | 29.29 | 21.47 | 33.12 | 28.50 | 15.41 | 12.34 | 37.64 | 32.88 | 33.78 | 28.60 |
| | O-Mem | **42.64** | **34.08** | **57.48** | **49.76** | 30.58 | 25.69 | **54.89** | **48.98** | **51.67** | **44.96** |
| | LangMemory** | 36.03 | 27.22 | 38.10 | 32.23 | 29.79 | 23.17 | 41.72 | 35.61 | 39.18 | 32.59 |
| | Mem0* | 17.19 | 12.06 | 3.59 | 3.37 | 12.24 | 8.57 | 12.74 | 10.62 | 11.62 | 9.24 |
| | ZEP** | 23.14 | 14.96 | 17.59 | 14.57 | 19.76 | 13.17 | 32.49 | 27.38 | 26.88 | 21.55 |
| GPT-4o-mini | MemoryOS** | 41.15 | 30.76 | 20.02 | 16.52 | **48.62** | **42.99** | 35.27 | 25.22 | 34.00 | 25.53 |
| | OpenAI** | 33.10 | 23.84 | 23.90 | 18.25 | 17.19 | 11.04 | 36.96 | 30.72 | 32.30 | 25.63 |
| | A-Mem** | 33.23 | 29.11 | 8.04 | 7.81 | 34.13 | 27.73 | 22.61 | 15.25 | 22.24 | 17.02 |
| | MEMOS** | 35.57 | 26.71 | 53.67 | **46.37** | 29.64 | 22.40 | 45.55 | 38.32 | 44.42 | 36.88 |
| | O-Mem | **44.17** | **34.78** | 53.54 | 45.65 | 25.24 | 19.22 | **54.53** | **48.33** | **50.60** | **43.48** |

* Mem0 was evaluated using its open-source version due to cost and accessibility. ** These results are cited from previous work.

Table 2: Performance comparison on PERSONAMEM with GPT-4.1.

| Method | Recall user shared facts | Suggest new ideas | Track full preference evolution | Revisit reasons behind preference updates | Provide preference-aligned recommendations | Generalize to new scenarios | Average |
|---|---|---|---|---|---|---|---|
| LangMemory | 31.29 | 24.73 | 53.24 | 81.82 | 40.00 | 8.77 | 42.61 |
| Mem0 | 32.13 | 15.05 | 54.68 | 80.81 | 52.73 | 57.89 | 46.86 |
| A-Mem | 63.01 | **27.96** | 54.68 | 85.86 | 69.09 | 57.89 | 59.42 |
| Memory OS | **72.72** | 17.20 | 58.27 | 78.79 | **72.72** | 56.14 | 58.74 |
| O-Mem | 67.81 | 21.51 | **61.15** | **89.90** | 65.45 | **73.68** | **62.99** |

Table 3: Performance comparison on Personalized Deep Research Bench with GPT-4.1*.

| Method | Goal Alignment | Content Alignment | Average |
|---|---|---|---|
| Mem0 | 37.32 | 35.54 | 36.43 |
| Memory OS | 40.60 | 39.67 | 40.14 |
| O-Mem | **44.69** | **44.29** | **44.49** |

* A fair comparison was conducted by generating all deep research reports through the centralized sonar-deep-research service (Son, 2025), leveraging retrievals from each method's individual memory system.

## 5 DISCUSSION

**Rethinking the Value of Memory Systems.** Do we truly need meticulously designed, complex memory systems? Most existing approaches adhere to a common paradigm: during retrieval, systems access processed user interactions rather than raw historical data. This design is largely driven by increasingly stringent privacy regulations worldwide (Voigt & Von dem Bussche, 2017; Calzada, 2022; Hosseini et al., 2024). By relying on abstracted user data, memory systems help AI companies mitigate legal risks while maintaining personalization capabilities. However, this abstraction comes at a significant cost: the irreversible loss of information fidelity and contextual nuance. For instance, a detailed user statement such as *"admiring a specific lamp and rug in a downtown antique store last Saturday"* may be compressed into a structured preference like *"[User] likes vintage home decor."* While efficient, this compression sacrifices granular details—specific objects, locations, and temporal context—that are crucial for precise and contextually relevant interactions.

To quantify this trade-off, we compare the performance of direct retrieval-augmented generation (RAG) over all raw interactions with O-Mem. As shown in Table 4, direct RAG achieves competitive performance (50.25 vs. 51.67 F1 score) despite its conceptual simplicity, though at a substantially higher computational cost (2.6K vs. 1.5K tokens). Notably, when compared to the results in Table 1, direct RAG with access to complete interaction history achieves competitive performance, highlighting the fundamental value of preserving raw interaction data. However, this comes at prohibitive computational costs that limit practical deployment. O-Mem addresses this critical limitation by achieving comparable performance with significantly reduced overhead, positioning it as a computationally efficient alternative that balances performance with practicality.

As depicted in Table 4, we also evaluate the practical deployment advantages of O-Mem by measuring average peak retrieval memory overhead (Liu et al., 2024) per response. O-Mem achieves a significant 30.6% reduction in peak memory overhead (from 33.16 MB to 22.99 MB), substantially relaxing hardware constraints for large-scale personalized inference. While direct RAG processes complete interaction histories verbatim, O-Mem

Table 4: Performance and efficiency comparison between direct retrieval from complete raw interaction history (Direct RAG) and O-Mem.

| Method | F1 (%) | Avg. Token Cost | Peak Memory Overhead (MB)* | Delay (s) |
|---|---|---|---|---|
| Direct RAG | 50.25 | 2.6K | 33.16 | 4.01 |
| O-Mem | 51.67 | 1.5K | 22.99 | 2.36 |

*For a fair comparison, the reported overhead for both RAG and O-Mem is calculated as the peak GPU memory usage minus the fixed memory allocated by the same embedding model used in our main experiments.

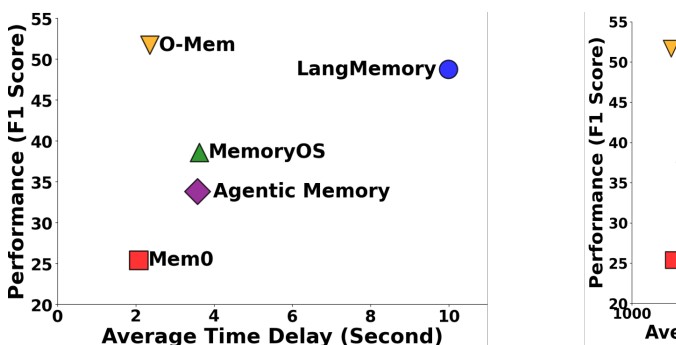 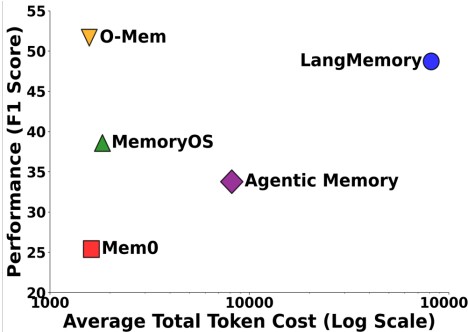

Figure 3: Trade-off between performance and efficiency of different memory frameworks. The left panel (a) compares the average latency per interaction. The MemoryOS latency was evaluated using FAISS-CPU due to compatibility issues on our computing platforms, thus representing a conservative estimate of its latency. (b) The right panel compares the average computational cost (in tokens) per interaction. Results demonstrate that O-Mem achieves a Pareto-optimal solution in both efficiency and overall performance.

maintains distilled representations that preserve semantic essence while drastically reducing sequence length. This design choice yields substantial computational memory benefits. For response latency, O-Mem demonstrates a 41.1% reduction in delay compared to direct RAG (from 4.01 seconds to 2.36 seconds), highlighting its efficiency for real-time applications.

**Efficiency Analysis.** We evaluate the efficiency of O-Mem by measuring the average token consumption and latency per response on the LoCoMo benchmark. The results, presented in Figure 3, substantiate that O-Mem achieves a superior balance between efficiency and effectiveness. Compared to the highest-performing baseline, LangMem (48.72 F1), O-Mem (51.67 F1) reduces token consumption by **94%** (from 80K to 1.5K tokens) and latency by **80%** (from 10.8s to 2.4s), while delivering superior performance. Against the **second-best performing** baseline, MemoryOS (38.58 F1), O-Mem not only secures a **34%** higher F1 score but also reduces latency by **34%** (2.4s vs. 3.6s). These results unequivocally demonstrate that O-Mem sets a new Pareto frontier for efficient and effective memory systems.

The efficiency advantage of O-Mem stems from two key design choices: The first is the independence of its retrieval operations across the three memory components. Unlike sequential architectures (e.g., A-men) that rely on a cascade of coarse-to-fine stages, O-Mem performs a one-time, concurrent retrieval across all three memory paths. Secondly, the retrieval mechanism of O-Mem utilizes user persona information, which, as opposed to raw user interaction records, typically contains less noise, thereby enhancing the cost-effectiveness of token usage. O-Mem also achieves a substantial reduction in **storage footprint**, requiring only nearly **3 MB per user**—**ten times less** than the almost **30 MB per user** consumed by Memory OS. This storage efficiency stems from our topic/keyword-based mapping design, which utilizes a lightweight index, in contrast to Memory OS which must store dense vector mappings for each memory chunk. Furthermore, O-Mem employs a radically simplified inference pipeline. Each response is generated through **only one** LLM invocation (three times for LangMemory). This streamlined workflow enables O-Mem to achieve superior efficiency with minimal reponse latency and computational expense.

**Memory Component Analysis** To quantify the contribution of each core module in our framework, we performed an ablation study on all three memory components—Persona Memory (PM), Episodic Memory (EM), and Working Memory (WM)—using the LocoMo benchmark. The results are summarized in Table 5. As indicated in the first three rows of the table, each module individually contributes to improved overall performance. However, **such performance gains could be partially attributed to the increased volume of retrieved information**, which leads to longer retrieval sequences and higher token consumption during response genera-

Table 5: Ablation study on different components of O-Mem using the LocoMo benchmark dataset.

| Memory Configuration | F1 (%) | Bleu-1 (%) | Total Tokens |
|---|---|---|---|
| WM only | 44.03 | 38.05 | 1.3K |
| WM + EM | 49.62 | 43.18 | 1.4K |
| WM + EM + PM | 51.67 | 44.96 | 1.5K |
| WM + EM (token-controlled) | 50.10 | 43.27 | 1.5K |
| WM only (token-controlled) | 46.07 | 39.95 | 1.5K |

tion. This trade-off between performance and efficiency has often been overlooked in prior ablation studies of memory-augmented systems. To definitively isolate this confound, we conducted a token-controlled ablation study (Rows 4–5 in Table 5), wherein the total token budget for each ablated configuration was fixed at 1.5K tokens to match that of the full O-Mem framework (WM+EM+PM). The clear performance gradient under a fixed token budget provides conclusive evidence that the performance gains are attributable to the *quality and relevance* of the information retrieved by each module, not merely to an increase in context. This finding confirms that each independent memory module of O-Mem effectively captures distinct and complementary aspects of the interactions.

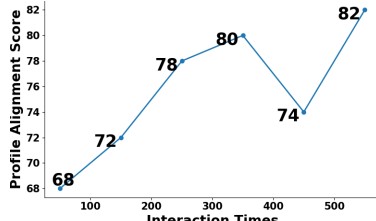

| Memory Configuration | Average Performance | Average Retrieval Length (Chars) |
|---|---|---|
| O-Mem | 44.49 | 6499 |
| O-Mem w/o Attributes | 42.14 | 28555 |

Table 6: Study concerning persona attribute on the Personalized Deep Research Bench dataset.

Figure 4: Memory Profile Alignment during Memory-Time Scaling

**Memory-Time Scaling for User Understanding.** We conduct a systematic evaluation of O-Mem's user understanding capability by examining how it **scales with the number of interactions** through two key analyses: (1) verifying the accuracy of persona attributes extracted from interaction data, and (2) assessing the practical utility of these attributes in personalizing the agent's responses. First, to evaluate the **scaling of extraction accuracy**, we collect persona attributes inferred by O-Mem from a single user's dialogue history **across increasing interaction counts**. These extracted attributes are then compared against the user's ground-truth profile using an LLM-as-judge scoring mechanism (Li et al., 2024a) to measure alignment (see Appendix for details). As shown in Figure 4, the extracted persona attributes gradually converge toward the ground-truth profile **as interactions scale**, demonstrating that O-Mem effectively refines its user understanding **through this scaling process**. Second, to measure the practical impact of persona attributes, we compare O-Mem with and without access to persona attributes on the Personalized Deep Research Bench dataset. Results in Table 6 show that incorporating persona attributes yields a significant improvement in response personalization (average performance increases from 42.14 to 44.49) while substantially reducing the retrieval length (from 28,555 to 6,499 characters). These results demonstrate that O-Mem's ability to extract and leverage user attributes **through scaled interactions** substantially enhances its performance in complex personalized text generation tasks, achieving stronger personalization with considerably improved efficiency.

## 6 CONCLUSION

In this paper, we propose O-Mem, a novel memory framework that enhances long-term human-AI interaction through dynamic user profiling and hierarchical memory retrieval. Unlike conventional approaches that rely solely on semantic retrieval of past messages, O-Mem actively constructs and refines user profiles from ongoing interactions. This approach effectively addresses the key limitations of conventional methods in maintaining long-term, consistent user context. Extensive experiments on three personalized benchmarks demonstrate that O-Mem achieves state-of-the-art performance while reducing token consumption by 94% and inference latency by 80% compared to its closest competitor, highlighting its superior efficiency. To support research in this area, we introduce Personalized Deep Research Bench, a new benchmark for personalized long-text generation. The proposed framework provides an effective solution for complex personalized text generation tasks, enabling LLM agents to deliver more coherent and contextually appropriate responses. Our work opens up promising directions for developing more efficient and human-like personalized AI assistants.

## 7 REPRODUCIBILITY STATEMENT

To facilitate the reproducibility of our work, we have provided comprehensive details throughout the manuscript. The proposed algorithm is elaborated in the Method section (Section 3), and the experimental configuration is detailed in the Experiments section (Section 4). Our experiments utilize the public LoCoMo and PERSONAMEM datasets, as well as the private Personalized Deep Research Bench dataset. The source code for our method and the Personalized Deep Research Bench dataset will be made publicly available upon the cancellation of anonymity.

## 8 ETHICS STATEMENT

We adhere to the ICLR Code of Ethics in this work. Our research aims to benefit society while upholding scientific rigor. We have considered potential risks to ensure the study is conducted ethically, fairly, and with respect for privacy and intellectual property. Furthermore, during the data construction process, we strictly followed the principle of informed consent, ensuring that all users were fully aware of the data's purpose and usage. Our data collection protocols were designed to safeguard their right to knowledge and privacy.

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

# 9 APPENDIX

## 9.1 USE OF LLMS

In this study, the large language model (LLM) was employed to assist in the writing process—specifically, to improve the articulation of experimental findings, perform proofreading for grammatical errors and typos, and verify mathematical formulations. It was not utilized for literature search or research ideation.

## 9.2 DETAILS CONCERNIG PERSONALIZED DEEP RESEARCH BENCH DATASET

### 9.2.1 CONSTRUCTION PROCESS

We recruited 25 demographically diverse volunteers (age, occupation, income, life stage) who, after training on data authenticity and privacy, mapped their personal information to a custom character model schema. To enrich these explicit profiles with dynamic context,we employed professional annotators simulate the daily interactions of these collected personas through the Xiaobu Memory APP to: (i) interact with an integrated assistant, and (ii) log natural-context memory fragments (e.g., travel, health, and family plans). This longitudinal data captured users' evolving interests, habits, and latent preferences. The benchmark's final stage required a principled pairing of profiles with memory tasks to generate personalized queries. Since random pairing would fail to capture user motivation and unique memory anchors, we employed a user-driven, committee-guided protocol. Each volunteer selected tasks from pool that resonated with personal memory traces (e.g., past milestones, domain knowledge, methodological hurdles). This ensured each task mirrored the participant's memory landscape, capturing idiosyncratic experiences and recall patterns. These 50 selected queries constitute the Personalized Deep Research Bench dataset for memory-based personalized deep research reports generation[1].

### 9.2.2 EVALUATION METRICS

To rigorously evaluate the efficacy of the generated personalized depth reports, we adopt a principled approach centered on two pivotal metrics: Goal Alignment and Content Alignment. These dimensions were selected because they directly correspond to the fundamental challenges of personalization: Goal Alignment assesses the system's capacity to comprehend and address a user's explicit and underlying motivations, while Content Alignment evaluates the appropriateness of the report's substantive content—its topic, depth, and breadth—to the user's unique knowledge background and interests. Our measurement methodology is grounded in the established LLM-as-Judge paradigm (Li et al., 2024a), for which we have designed structured prompts to operationalize these indicators. While we acknowledge that no evaluation framework can claim perfect objectivity, we contend that this approach provides a systematic, transparent, and currently optimal approximation for assessing the nuanced construct of personalization quality.

### 9.2.3 PERSONA ALIGNMENT SCORE

To evaluate the fidelity of the user profiles extracted by O-Mem, we have designed structured prompts that direct a large language model to compare the extracted profiles against the ground-truth user profiles and assign a consistency score. Similar to the evaluation of goal and content alignment, this approach leverages the LLM-as-a-judge paradigm. We acknowledge the inherent limitations of this method in achieving perfect objectivity. Its primary objective, however, is not to provide a precise measurement but to fundamentally assess whether O-Mem can construct a dynamic and increasingly accurate user portrait as the number of interactions scales.

### 9.2.4 EVALUATION PROMPT

> **Prompt for Goal Alignment Criteria Generation**
>
> You are an experienced research article evaluation expert. You excel at breaking down abstract evaluation dimensions (such as "Goal Understanding and Personalization Insight") into actionable, clear evaluation criteria tailored to the specific research task and user persona, and assigning reasonable weights with explanations for each criterion.
> `</system_role>`

---

[1]The full Personalized Deep Research Bench benchmark released to the community will comprise the entire initial query set, including the 50 highly personalized queries selected for this study as well as the broader collection, which, despite being less discriminative for memory personalization, remains a valuable asset for future research.

```
<user_prompt>
```
**Background**: We are evaluating a research article written for the following research task under the dimension of Goal Alignment.

**Goal Alignment:** Whether the research fully and accurately understands the relationship between the task and the user persona, extracts deep and implicit needs, and generates a personalized report based on that understanding, with a focus on performing user-centered, deeply personalized matching between the user persona and task requirements.

```
<task>
```
"{task_prompt}"
```
</task>
```

The user persona is as follows:
```
<persona>
```
"{persona_prompt}"
```
</persona>
```

```
<instruction>
```
Your goal:

For the Goal Alignment dimension of this research article, formulate a set of detailed, specific, and highly targeted evaluation criteria that are tightly aligned with the above `<task>` and `<persona>`. You need to:

1. Deeply analyze the user persona and task scenario: Thoroughly examine the background characteristics, knowledge structure, cognitive habits, and latent expectations of `<persona>`. Combine this with the specific application scenario of `<task>` to identify the user's core explicit needs and deeper implicit needs.

2. Formulate personalized evaluation criteria: Based on the above analysis, propose specific evaluation criteria that reflect a deep understanding of `<persona>` and a close fit to the `<task>` scenario. These criteria should assess whether the content is well adapted to the user persona in style, depth, perspective, and practicality.

3. Explain the personalization rationale: Provide a brief explanation (explanation) for each criterion, clarifying how it addresses the specific attributes of `<persona>` or special requirements of `<task>`, and why such targeting is critical to achieving a good match.

4. Assign rational weights: Assign a weight (weight) to each criterion, ensuring that the total sum is 1.0. The distribution of weights should directly reflect the relative importance of each criterion in measuring how well the content matches "this particular user" in "this particular task." The closer a criterion is tied to persona characteristics and task scenario, the higher its weight should be.

Core requirements:

1. Deep personalization orientation: The analysis, criteria, explanations, and weights must be deeply rooted in the uniqueness of `<persona>` (e.g., their professional background, cognitive level, decision-making preferences, emotional needs) and the specific context of `<task>`. Avoid generic or templated evaluation.

2. Focus on contextual responsiveness and resonance: The criteria should evaluate whether the content not only responds to the task at the informational level but also resonates with the context and expectations implied by the user persona in terms of expression style, reasoning logic, case selection, and level of detail.

3. Rationale must reflect targeting: The `<analysis>` section must clearly explain how key features were extracted from the given `<persona>` and `<task>` to form these personalized criteria. Each criterion's explanation must directly show how it serves this specific user and task.

4. Weights must reflect personalization priorities: The weight distribution must logically demonstrate which aspects of alignment are the most critical success factors for "this user" completing "this task."

5. Standard output format: Strictly follow the example format below. First output the `<analysis>` text, then immediately provide the `<json_output>`.

```
</instruction>
```

```
<example_rational>
```
The example below demonstrates **how to develop Goal Alignment evaluation criteria based on the task requirements**. Focus on understanding the **thinking process and analytical approach** used in

the example, rather than simply copying its content or numerical weights.
</example_rational>
...

Please strictly follow the above instructions and methodology. Now, for the following specific task, start your work:
<task>
"{task_prompt}"
</task>

<persona>
"{persona_prompt}"
</persona>

Please output your <analysis> and <json_output>.
</user_prompt>

---

### Prompt for Content Alignment Criteria Generation

You are an experienced research article evaluation expert. You are skilled at breaking down abstract evaluation dimensions (such as "Content Alignment") into actionable, clear, and specific evaluation criteria tailored to the given research task and user persona, and assigning reasonable weights and explanations for each criterion.
</system_role>
<user_prompt>
**Background**: We are providing a personalized scoring rubric for a specific task and user persona from the dimension of **Content Alignment**.
**Content Alignment**: Whether the research content is customized based on the user's interests, knowledge background, and other preferences.

<task>
"{task_prompt}"
</task>

The user persona is as follows:
<persona>
"{persona_prompt}"
</persona>
<instruction>
**Your Goal**: For the **Content Alignment** dimension of this research article, create a set of detailed, concrete, and highly tailored evaluation criteria for the above <task> and <persona>. You need to:

1. **Analyze the Task and Persona**: Deeply analyze <task> and <persona> to infer the user's potential interests, knowledge background, and the depth and breadth of content they may prefer.

2. **Formulate Criteria**: Based on your analysis, propose specific evaluation criteria that focus on whether the report's content matches the user's interest points and knowledge level.

3. **Provide Explanations**: For each criterion, provide a brief explanation (`explanation`) explaining why it is important for evaluating the content alignment for this <task>.

4. **Assign Weights**: Assign a reasonable weight to each criterion (`weight`), ensuring that the sum of all weights equals exactly 1.0. The weight allocation should logically reflect the personalization-first principle: criteria directly tied to unique personal traits, exclusive preferences, or specific contextual needs in the user persona should receive higher weights, as they are key to achieving true personalized content alignment.

5. **Avoid Overlap**: Make sure the evaluation criteria focus solely on the **Content Alignment** dimension, avoiding overlap with other dimensions such as Goal Alignment, Expression Style Alignment, and Practicality/Actionability.

**Core Requirements**:

1. **Strongly Linked to the Persona**: The analysis, criteria, explanations, and weights must be directly connected to the user's interests, knowledge background, or content preferences.

2. **Focus on Content Selection and Depth**: The criteria should assess whether the choice of content is precise and whether the depth is appropriate, rather than merely evaluating whether information is presented.

3. **Provide Sufficient Rationale**: The <analysis> section must clearly articulate the overall reasoning behind formulating these criteria and weights, linking them to <task> and <persona>. Each `explanation` must clarify why the individual criterion is relevant.

4. **Reasonable Weighting**: The weight distribution should be logical, reflecting the relative importance of each criterion in measuring content alignment, with particular emphasis on giving higher priority to personalized aspects.

5. **Standardized Output Format**: Strictly follow the format below — output the <analysis> text first, immediately followed by <json_output>.

</instruction>
<example_rational>
The following example demonstrates **how to formulate content alignment evaluation criteria based on the task requirements and user persona**. Pay close attention to the **thinking process and analytical approach** in this example, rather than simply copying the content or weight values.
</example_rational>
. . .
Please strictly follow the above instructions and methodology. Now, for the following specific task, start your work:
<task>
"{task_prompt}"
</task>
<persona>
"{persona_prompt}"
</persona>
Please output your <analysis> and <json_output>.
</user_prompt>

---

Scoring Prompt for Personalization

(For convenience and under time constraints, a temporary, unrefined prompt was employed for scoring during the experiment. The additional personalized indicators included in this temporary prompt—beyond the core metrics of goal alignment and content alignment—were ultimately discarded due to conceptual overlap. Therefore, this provisional prompt is functionally equivalent to our intended final evaluation design.)

<system_role>You are a strict, meticulous, and objective expert in evaluating personalized research articles. You excel at deeply evaluating research articles based on specific personalization assessment criteria, providing precise scores and clear justifications.</system_role>

<user_prompt>
**Task Background**
You are given an in-depth research task. Your job is to evaluate a research article written for this task in terms of its performance in **Personalization Alignment**. We will evaluate it across the following four dimensions:

1. Goal Alignment

2. Content Alignment

3. Presentation Fit

4. Actionability & Practicality

<task>
{task_prompt}
</task>

**User Persona**
<persona>
{persona_prompt}
</persona>

**Article to be Evaluated**
<target_article>
{article}
</target_article>

**Evaluation Criteria**
You must evaluate the specific performance of this article in terms of personalization alignment, **following the criteria list below**, outputting your analysis and then assigning a score from 0–10. Each criterion includes its explanation, which you should read carefully.
<criteria_list>
{criteria_list}
</criteria_list>

<Instruction>
**Your Task**
Strictly follow **each criterion** in <criteria_list> to evaluate how <target_article> meets that criterion. You must:

1. **Analyze Each Criterion**: For each item in the list, think about how the article meets the requirements of that criterion.

2. **Analytical Evaluation**: Combine the article content, the task, and the user persona to analyze the article's performance for that criterion, pointing out both strengths and weaknesses.

3. **Scoring**: Based on your analysis, give a score between 0 and 10 (integer) for the article's performance on that criterion.

**Scoring Rules**
For each criterion, give a score between 0 and 10 (integer). The score should reflect the quality of the article's performance:

- 0–2 points: Very poor. Almost completely fails to meet the requirement.

- 2–4 points: Poor. Meets the requirement only partially, with significant shortcomings.

- 4–6 points: Average. Basically meets the requirement; neither particularly good nor bad.

- 6–8 points: Good. Mostly meets the requirement, with notable strengths.

- 8–10 points: Excellent/Outstanding. Fully or exceptionally meets the requirement.

**Output Format Requirements**
Strictly follow the <output_format> below to output the evaluation results for **each criterion**. **Do not include any irrelevant content, introductions, or conclusions**. Start from the first dimension and output all dimensions and their criteria in sequence:
</Instruction>

<output_format>

```
{
    "goal_alignment": [
        {
            "criterion": "[The text of the first
            Goal Alignment criterion]",
            "analysis": "[Analysis]",
            "target_score": "[integer score 0-10]"
        },
        {
            "criterion": "[The text of the second
            Goal Alignment criterion]",
            "analysis": "[Analysis]",
            "target_score": "[integer score 0-10]"
        },
        ...
    ],
    "content_alignment": [
        {
            "criterion": "[The text of the first Content Alignment
            criterion]",
```

```
21          "analysis": "[Analysis]",
22          "target_score": "[integer score 0-10]"
23      },
24      ...
25  ],
26  "presentation_fit": [
27      {
28          "criterion": "[The text of the first Presentation Fit
29          criterion]",
30          "analysis": "[Analysis]",
31          "target_score": "[integer score 0-10]"
32      },
33      ...
34  ],
35  "actionability_practicality": [
36      {
37          "criterion": "[The text of the first Actionability
38          & Practicality criterion]",
39          "analysis": "[Analysis]",
40          "target_score": "[integer score 0-10]"
41      },
42      ...
43  ]
44 }
```

</output_format>

---

**Prompt for Persona Align Score**

<system_role>
You are an experienced user research expert, skilled in analyzing and comparing user personas. Your task is to carefully compare a "Preset User Persona" and a "System Dynamically Learned User Persona", and identify the key similarities and differences between them.
</system_role>

<user_prompt>
**Your analysis must strictly follow these four dimensions:**

1. **Basic Attributes & Goals**: Compare similarities and differences in areas such as occupation, identity, core objectives, and usage motivations.

2. **Behavioral Patterns**: Compare similarities and differences in areas such as usage frequency, commonly used features, and interaction depth.

3. **Needs & Preferences**: Compare similarities and differences in areas such as content preferences, feature requirements, and pain points.

4. **Overall Image Differences**: Summarize the overall perceptual differences between the two personas (e.g., "Diligent Learner" vs. "Efficient Problem Solver").

Input Data:
- **Preset User Persona**: $preset_{persona_t}ext$
- **System Learned User Persona**: $learned_{persona_t}ext$
Output Requirements:
Please output your analysis results in **pure JSON format** only, without any additional explanations.
The JSON structure should be as follows:

```
{{
  "comparison_by_dimension": {{
    "Basic Attributes & Goals": {{
      "Preset Persona Summary": "one-sentence summary",
      "Learned Persona Summary": "one-sentence summary",
      "Key Similarities": ["point 1", "point 2", ...],
      "Key Differences": ["point 1", "point 2", ...],
      "Difference Level": "High/Medium/Low"
      // Judge based on the significance of differences
```

```
    }},
    "Behavioral Patterns": {{
       ... // Same structure as above
    }},
    "Needs & Preferences": {{
       ... // Same structure as above
    }}
  }},
  "overall_summary": {{
    "Preset Persona Overall Image": "a descriptive label or phrase",
    "Learned Persona Overall Image": "a descriptive label or phrase",
    "Overall Alignment Score": "integer from 0-100", // 100
    indicates complete alignment
    "Most Important Insight": "one or two sentences explaining the
    most critical insight"
  }}
}}

</user_prompt>
```

## 9.3  INTERACTION UNDERSTANDING PROMPT

### UNDERSTAND USER EXPERIENCE PROMPT

Perform topic tagging on this message from user following these rules:
1. Generate machine-readable tag
2. Tag should cover:
- Only one primary event concerning the user messages.
- The author's attitude towards the event.
- The topic should be the subject of the message which the user held attitude towards.
- The topic and reason behind the attitude, sometimes you need to infer the attitude from the users' words.
- The facts or events infered or revealed from the user's message.
- If the author mention the time of the facts or events, the tag should also include the time inferred from the message (e.g., last day, last week)
- Any attributes of the user revealed by the user's message (e.g., demographic features,biographical information,etc).
3. Use this JSON format:

```
{{
"text": "original message",
"tags": {{
"topic": ["event"],
"attitude": ["attitude towards the event":Postive or Negative or Mixed]
"reason" :["The reason concenring the attitude towards the event"]
"facts": ["The facts or events infered from the user's message"]
"attributes": ["The attributes of the user revealed by
the user's message"]
}},
"summary": "One sentence summary of the message"
"rationale": "brief explanation concenring why raising these tags"
}}
```

Example Input: "The jazz workshop helped me overcome performance anxiety"
Example Output:

```
{{
"text": "Last week's jazz workshop helped me overcome
performance anxiety since the tutors are so patients.",
"tags": {{
    "topic": ["music workshop"],
    "attitude": ["Positive"],
    "reason": ["The tutors can teach the use patiently."],
    "facts":["join jazz workshop last week"],
```

```
    "attributes": ["user worrys about jazz performance"]
}},
"summary": "Jazz workshop helped the user overcome
performance anxiety."
"rationale": "The user's performance anxiety was
alleviated with the help of Jazz Workshop.
Therefore , he is positive towards Jazz Workshop."
}}
```

Example Input: "I stop playing basketball for this semester due to too much stress."
Example Output:

```
{{
    "text": "The user step away from playing baskerball
    due to too much stress.",
    "tags": {{
        "topic": ["playing basketball"],
        "attitude": ["negative"],
        "reason": ["Too much stree for playing basketball"],
        "facts:["stop playing basketball"],
        "attributes": ["user hate stress"]
    }},
    "summary": "The user stop playing baskerball due to
    too much stress."
    "rationale": "The user stop playing baskerball due
    to too much stress.
    Therefore, the user is negative towards playing basketball."
}}
```

Example Input: "I go back to play basketball due to strenghten my body yesterday."
Example Output:

```
{{
"text": "The user return to play basketball due to
strenghten the body.",
"tags": {{
    "topic": ["playing basketball"],
    "attitude": ["Positive"],
    "reason": ["Baskterball could help strenghtening the body"],
    "facts":["return to play basketball yesterday"],
    "attributes": ["User value the body"]
}},
"summary": "The user go back to play basketball due to
strenghten the body."
"rationale": "he user go back to play basketball due to
strenghten the body. There, the user is positive towards
playing basketball."
}}
```

Example Input: "I hate playing basktetball due to its preasure"
Example Output:

```
{{
"text": "I hate playing basktetball since I move from my
hometowm GuangZhou due to its preasure.",
"tags": {{
    "topic": ["hate playing basketball"],
    "attitude": ["negative"],
    "reason": ["The user hates playing basktetball for preasure."],
    "facts":["hate playing basketball"],
    "attributes": ["user hate stress","user's hometown
    is GuangZhou"]
}},
"summary": "The user go back to play basketball due to
strenghten the body."
```

```
"rationale": "The user go back to play basketball due to
strenghten the body. There, the user is positive towards
playing basketball."
}}
```

Now analyze this message: "message"

