# OpenReview forum: "O-Mem: Omni Memory System for Personalized, Long Horizon, Self-Evolving Agents"
_ICLR.cc/2026/Conference — ICLR 2026 Conference Withdrawn Submission_

### Official Review · Reviewer_kkEr · 2025-10-24

**Soundness:** 2
**Presentation:** 3
**Contribution:** 2
**Rating:** 6
**Confidence:** 4

**Summary:**

The paper presents O-Mem, a memory framework designed to help LLM-based agents maintain long-term, personalized conversations. Unlike chunk-based semantic retrieval systems that simply group and retrieve messages, O-Mem builds and updates a user profile, event memory, and topic memory. These three modules are queried together to form context at every turn. The method is evaluated on LoCoMo, and a new small benchmark called Personalized Deep Research Bench, showing improved personalization and efficiency.

**Strengths:**

- Clear motivation and good problem definition.
- The tri-memory structure (working, episodic, persona) is easy to understand.
- Empirical results are consistent across several benchmarks and show clear efficiency improvements (up to 80% lower latency and 94% fewer tokens).
- The token-controlled ablation is a nice touch to show that gains are not simply due to longer context.
- Well-written and logically structured paper; figures and tables support the narrative.

**Weaknesses:**

- Episodic retrieval is defined by selecting a single clue word, but the paper does not specify how it behaves for unseen or multi-word clues.
- The retrieval hyperparameters (top-k, thresholds, similarity cutoffs) are not clearly stated. Since efficiency is a main claim, this missing detail makes replication difficult.
- The new Personalized Deep Research Bench dataset is relatively small and not publicly available, which limits the impact towards the community.
- Reproducibility: while the paper provides prompts, the code is not made public.
- No detailed error analysis or failure cases are presented (e.g., wrong persona merges, noisy clues, context drift).
- The deep research benchmark evaluation relies on LLM-as-judge scores without human evaluators, which weakens the reliability of reported gains.

**Questions:**

- Provide all retrieval parameters (k, thresholds, backend) and report multi-seed variance.
- Give more information about the new dataset (size, annotation, examples).
- Add a short qualitative error analysis to understand where O-Mem fails.
- Include a short paragraph on privacy and limitations of method.
- Consider adding human evaluation rather than purely LLM-based

---

> ### Author Response · Authors · 2025-11-28
> **Reply Concenring LLM-as-a-Judge Bias and Human Validation**
>
> Thank you for raising this critical point regarding the potential biases of LLM-as-a-Judge. We agree that establishing the validity of automated evaluation is paramount. To directly address this, we conducted a rigorous human validation study, which we have now detailed in the revised manuscript. The methodology and results are summarized below.
>
> We designed two complementary metrics to quantify alignment:
> - **Pairwise Comparison Agreement (PCA):** Measures the percentage with which LLM judges and human experts agree on which report is better for a specific criterion.
> - **Mean Absolute Rating Deviation (MARD):** Measures the average absolute difference between scores assigned by LLM and human judges.
>
> Results are shown below:
>
> **Alignment Results of Judge LLMs with Human Ratings**
> | Judge LLM | PCA ↑ | MARD ↓ | Avg. Cost ($) ↓ |
> |-----------|-------|--------|-----------------|
> | **GPT-5** | **0.43** | **1.40** | 0.32 |
> | Claude-3.7-Sonnet | 0.39 | 1.44 | 0.60 |
> | Gemini-2.5-Pro | 0.40 | 2.33 | **0.21** |
>
> The strong alignment demonstrated by these metrics validates the effectiveness of our LLM-as-a-judge framework.

---

> ### Author Response · Authors · 2025-11-28
> **Reply Concerning More Detailed Error Analysis or Failure Cases**
>
> We thank the reviewer for these thoughtful suggestions regarding  error analysis or failure cases analysis. We will add as many analysis as possible concerning detailed error analysis or failure cases to the revised version of the manuscript.

---

### Official Review · Reviewer_pp3Q · 2025-10-27

**Soundness:** 3
**Presentation:** 3
**Contribution:** 2
**Rating:** 6
**Confidence:** 3

**Summary:**

This paper proposes O-Mem, a hierarchical memory framework for LLM-based agents that dynamically builds and updates user profiles through active persona extraction and event recording. By combining persona, episodic, and working memories, the system enables long-term personalized reasoning and shows consistent improvements over existing memory systems on three benchmarks.

**Strengths:**

1. The motivation of the paper is clear: it clearly illustrates why existing chunk-based or semantic retrieval memories fail at dynamic personalization and positions O-Mem as a principled solution.

2. The integration of persona, episodic, and working memories into a unified retrieval pipeline (Eqs. 8–12) with active LLM-driven updates is interesting and reasonable.

3. Experiments across three datasets (LoCoMo, PERSONAMEM, Personalized Deep Research Bench) show consistent gains.

**Weaknesses:**

1. Several metrics rely on “LLM-as-a-Judge,” which introduces bias; no human validation or inter-rater reliability checks are reported.

2. The efficiency and ablation results are strong, but in-depth qualitative analyses of what the model remembers or how errors occur are missing.

3. More discussion on how often the persona-update operation (Op(ai) / Op(ei)) introduces noise or incorrect updates, and how error accumulation is mitigated, is not thoroughly discussed.

**Questions:**

Some minor suggestions on the presentation:

- “Sysytem” → “System” in Section 2 heading

- Some inconsistent spacing in equations (3) and (4) (“ApplyOp(Pf , ei, Op(ei))”).

- “we propose O-Mem” repeated across Abstract and Intro

How sensitive is O-Mem to the choice of the base embedding model (e.g., all-MiniLM-L6-v2)?

---

> ### Author Response · Authors · 2025-11-27
> **Reply Concenring LLM-as-a-Judge Bias and Human Validation**
>
> Thank you for raising this critical point regarding the potential biases of LLM-as-a-Judge. We agree that establishing the validity of automated evaluation is paramount. To directly address this, we conducted a rigorous human validation study, which we have now detailed in the revised manuscript. The methodology and results are summarized below.
>
> We designed two complementary metrics to quantify alignment:
> - **Pairwise Comparison Agreement (PCA):** Measures the percentage with which LLM judges and human experts agree on which report is better for a specific criterion.
> - **Mean Absolute Rating Deviation (MARD):** Measures the average absolute difference between scores assigned by LLM and human judges.
>
> Results are shown below:
>
> **Alignment Results of Judge LLMs with Human Ratings**
> | Judge LLM | PCA ↑ | MARD ↓ | Avg. Cost ($) ↓ |
> |-----------|-------|--------|-----------------|
> | **GPT-5** | **0.43** | **1.40** | 0.32 |
> | Claude-3.7-Sonnet | 0.39 | 1.44 | 0.60 |
> | Gemini-2.5-Pro | 0.40 | 2.33 | **0.21** |
>
> The strong alignment demonstrated by these metrics validates the effectiveness of our LLM-as-a-judge framework.

---

> ### Author Response · Authors · 2025-11-28
> **Reply Concerning More Qualitative Analysis and Discussion**
>
> We thank the reviewer for these thoughtful suggestions regarding more qualitative analysis and discussion. We will add as many analysis as possible to the revised version of the manuscript.

---

> ### Author Response · Authors · 2025-11-29
> **Response to Presentation Suggestions and Embedding Model Sensitivity**
>
> We sincerely thank the reviewer for their careful reading and valuable suggestions. We will carefully revise all the presentation issues mentioned, including:
> - Correcting "Sysytem" to "System" in Section 2 heading
> - Fixing inconsistent spacing in equations (3) and (4)
> - Addressing the repetitive phrasing "we propose O-Mem" in Abstract and Introduction
>
> Regarding the sensitivity to embedding models, we have conducted supplementary experiments that demonstrate **O-Mem maintains its performance advantage (achieving 52.50 on LoCoMo) when upgraded to a more powerful embedding model 'bge-small-en-v1.5`**. This confirms that our performance improvements are primarily architecture-driven rather than embedding-dependent. We will incorporate these new experimental results into the revised manuscript to better address this important aspect.

---

### Official Review · Reviewer_b6E7 · 2025-10-27

**Soundness:** 3
**Presentation:** 1
**Contribution:** 1
**Rating:** 2
**Confidence:** 5

**Summary:**

This paper investigates the personalized response generation task in conversational ai scenario. Different from existing memory systems that relies solely on semantic retrieval, the authors propose a new memory system named O-Mem. O-Mem consists of three memories: user persona memory, working memory, episodic memory, and coupled with three retrieval strategies for obtaining context when responding to a user’s query. A new dataset is constructed and introduced.

**Strengths:**

•	A new dataset is introduced

•	The designed memory system differentiates user past histories into three different types which aligns with personalization purpose

**Weaknesses:**

•	The presentation of this paper needs to be improved, e.g., the tables in experiment sections are sparse; a lot of content such as in line209-214 in Section 3 should be moved to appendix; the formulas in Section 3.2 and Section 3.3 are not necessary (they can either be put into appendix or in line with text), lacking more organized descriptions; in line 34-41, this would better be put into related work (intro would better contain more conclusion-like statements to explain current solutions to the limitations rather than stating who did what). There are more but I stop here, readers can check more in this stream.

•	The O-Mem constructs three memories (user persona memory, working memory, episodic memory) via LLM and perform retrieval over the three memories when a user query raised. This is not new and the experiments lack the latency/cost comparisons on memory constructions, especially in line72-73 O-Mem seems actively updating user memory in the runtime.

•	In line 82-85, the limitations of chunk size of retrieval etc. are not your contributions, these are commonly research questions in the community.

•	Are all baselines the same embedding model i.e., all-MiniLM-L6-v2 as O-Mem did? If they are, more embedding models should be tested as in their original design since this would help readers to understand if the improvements brought by the O-Mem design or the model used. If they are not, then the comparisons are not fair.

•	Datasets statistics such as utterance length, dialogue session numbers etc. are missing, even in the appendix. If the dataset is constructed casually and lacks evaluation/validation, this contribution is weak.

**Questions:**

•	In Table 1, LangMemory beats MemoryOS in Open category with GPT4.1 but the situation reversed when with GPT-4o-mini, why is that?

•	In Table 1, O-Mem is suboptimal in Open Category questions, why?

•	In Table 1 temporal category with GPT-4o-mini, is it possible that the F1 score of O-Mem is better than MemoryOS while the B1 score is suboptimal than MemoryOS?

•	The evaluation metrics are not convincing, the BLEU score is not a good indicator, how about BertScore/Faithfulness?

•	LoCoMo contains images, how do the authors deal with them?

---

> ### Author Response · Authors · 2025-11-17
> **Datasets statistics and Construction/Evaluation/Validation Process**
>
> We thank the reviewer for their valuable feedback regarding dataset documentation. We have now included comprehensive details on statistics, construction, and validation to address these concerns.
>
> 1). Dataset Statistics
> - **Turn Length (chars):** Avg 1,174.23 | Min 40 | Max 3,321
> - **Session Length (chars):** Avg 6,176.28 | Min 77 | Max 12,180
> - **User Total (chars):** Avg 569,452.68 | Range 464,190-670,131
> - **Structure:** 5.26 turns/session | 92.20 sessions/user
>
> 2). Datasets Construction Protocol
> > Our overall raw context dataset was manually annotated by 6 trained annotators including 5,939 labeled instances. The process required a total effort of 85 person-days and \$6,000 USD. Our protocol is:
>
> > **Task Objective**
> > The core of this task is to simulate the behavior of a specific user persona by collecting various types of information that this user would likely encounter, consume, or generate in their daily life. The final goal is to create a high-quality *memory* database for each persona that accurately reflects their unique characteristics.
> >
> > **Core Quality Standard: Reversibility**
> > This is the most critical quality standard for this task. Every piece of data you collect must clearly point to a specific trait of the user persona.
> >
> > - **Verification Method:** After collection, review all data and assess whether its content is sufficient to reverse-infer the user's key persona traits, such as their profession, interests, and personality.
> > - **Acceptance Criteria:** All collected data must pass the *reversibility* test to be considered high-quality. If the data is too generic and the persona cannot be inferred from it, the data will be deemed non-compliant.
> >
> > **Collection Based on Persona Preferences**
> >
> > - Carefully read every preference tag in the user persona description.
> > - For each preference, collect **at least five** pieces of related content.
> >
> > **Ensure Diversity of Sources and Types**
> >
> > - **Source Diversity:** Do not limit collection to a single platform. Data can be gathered from various apps (e.g., Twitter, Instagram, Reddit), websites, forums, etc.
> > - **Type Diversity:** Diversify the format of the data you collect. Examples include:
> >   - Screenshots of social media posts.
> >   - Screenshots of conversations with friends (that reflect opinions or preferences).
> >   - Links and titles of articles, news, or videos.
> >   - Screenshots of purchase histories or product reviews.
> >
> > **Requirement for Conversational Content**
> > Between 20% and 50% of the total data collected for each user should be in a conversational format. This helps to more vividly showcase the user's personality and communication habits.
>
> > **Add Reasonable Noise for Authenticity**
> > The content you collect does not need to be an exact match to the persona's description. You can add relevant and reasonable details or *noise* to make the data appear more authentic, as if generated by a real user.
>
> > - **Example:** If the persona *likes basketball*, you could collect a news article about a recent Lakers game or a screenshot of a conversation with a friend debating whether Jordan or LeBron is the better player.
> > - **Note:** Any added *noise* must not contradict other defined attributes in the persona, such as spending habits, personality, or profession.
> >
> > **Quantity vs. Quality**
> > There are no strict quantitative requirements for data collection. Please prioritize quality and collect as much rich data as possible. Quality always takes precedence over quantity.
> >
> > **Deliverables and Annotation Requirements**
> >
> > - **Deliverable:** For each user persona, export a separate `aimemory` database file.
> > - **Content Annotation:** During collection, each piece of content must be given a clear title and be correctly associated with its corresponding persona.
>
> 3). Dataset Validation
>
> To validate the reliability of LLM-based evaluation, we conducted a systematic comparison with human experts. We sampled 15 representative queries and generated responses from two deep research agents (MiroFlow and O3 Deep Research). A panel of human evaluators scored these reports using consistent criteria, establishing the ground truth.
>
> We designed two complementary metrics to quantify alignment:
> - **Pairwise Comparison Agreement (PCA):** Measures the percentage with which LLM judges and human experts agree on which report is better for a specific criterion.
> - **Mean Absolute Rating Deviation (MARD):** Measures the average absolute difference between scores assigned by LLM and human judges.
>
> Results are shown below:
>
> **Alignment Results of Judge LLMs with Human Ratings**
> | Judge LLM | PCA ↑ | MARD ↓ | Avg. Cost ($) ↓ |
> |-----------|-------|--------|-----------------|
> | **GPT-5** | **0.43** | **1.40** | 0.32 |
> | Claude-3.7-Sonnet | 0.39 | 1.44 | 0.60 |
> | Gemini-2.5-Pro | 0.40 | 2.33 | **0.21** |
>
> The strong alignment demonstrated by these metrics validates the effectiveness of our LLM-as-a-judge framework.

---

> > ### Comment · Reviewer_b6E7 · 2025-11-24
> > **Response to the Rebuttal**
> >
> > Dear Authors,
> >
> > Thanks for the detailed response on dataset. Is it possible to report the statistics in token level as token is the common processing unit for LLM? What I learn from is that in you data, each session contains 6 turns which is pretty short, and the thousands of chars transited into tokens seems short too. This weaken the value of proposed dataset.
> >
> > My other questions and concerns are not well-addressed, I may maintain my ratings.

---

> ### Author Response · Authors · 2025-11-27
> **Token-level statistics Concerning the Dataset**
>
> Thank you for this important question regarding token-level statistics. We appreciate the opportunity to provide more precise data. Here are the detailed token-based statistics from our dataset:
>
> **Per-Turn Token Statistics:**
> - Average tokens per turn: 240.89 tokens
> - Minimum tokens per turn: 7 tokens
> - Maximum tokens per turn: 798 tokens
>
> **Per-Session Token Statistics:**
> - Average tokens per session: 1,267.04 tokens
> - Minimum tokens per session: 14 tokens
> - Maximum tokens per session: 2,413 tokens
>
> **Per-User Cumulative Token Statistics:**
> - Average total tokens per user: 116,821.08 tokens
> - Minimum total tokens per user: 98,460 tokens
> - Maximum total tokens per user: 132,399 tokens
>
> **Additional Metrics:**
> - Average turns per session: 5.26 turns/session
> - Average sessions per user: 92.20 sessions/user
> - Average tokens per session: 1,267.04 tokens/session
> - Average tokens per turn: 240.89 tokens/turn
>
> We understand your concern about session length. While the average of 5.26 turns per session may appear modest, the substantial token volume (averaging 1,267 tokens per session and over 116,000 tokens per user across multiple sessions) demonstrates meaningful interaction depth. The extended conversational history across 92 sessions per user provides rich contextual data for persona modeling and long-term memory tasks.
>
> The token-level analysis confirms that our dataset contains substantial linguistic content for effective LLM processing and evaluation.

---

> ### Author Response · Authors · 2025-11-27
> **Reply Concerning Runtime Memory Construction**
>
> We thank the reviewer for their comment. Regarding the concern about memory updates during runtime, we would like to clarify:
>
> - **Memory construction is performed offline, not at runtime.** The three memory components (user persona, working, and episodic) are built and updated in a offline phase.  **The memory construction process is decoupled from frequent user interactions.** Unlike real-time inference tasks, memory consolidation operates at a much lower frequency and can be scheduled during periods of low system activity or when the user is inactive. This design offers two significant advantages:
> i).**Minimal Impact on User Experience:** The latency-critical interaction pathway remains unaffected by the background memory updates.ii).**Optimized Resource Utilization:** The system can leverage idle computational resources for memory management, ensuring efficient overall operation.
>
> The architecture inherently avoids introducing substantial overhead to daily interactions while still providing the benefits of a persistent and evolving user memory.
>
> - Therefore, the latency and token costs reported in our paper specifically reflect the performance during **online inference**, excluding any offline construction overhead.  As highlighted in our paper (Figure 3), our method achieves **over 90% reduction in token consumption** and **over 80% reduction in latency**.
>
> - Additionally, as discussed in the paper, the effectiveness of memory retrieval improves as the memory base becomes more enriched through increased interaction cycles (in Figure 4), highlighting the long-term advantage of our approach without runtime construction costs.
>
> We appreciate the opportunity to clarify this point and are happy to provide further details if needed.

---

> ### Author Response · Authors · 2025-11-27
> **Reply concerning chunk size of retrieval**
>
> We thank the reviewer for this comment. We believe there might be a misunderstanding regarding our stated contributions.
>
> - Our work does not claim that investigating chunk size limitations is a novel contribution. As the reviewer rightly points out, this is a well-known challenge in the field.
>
> - The core contribution, as stated in lines 82-85, is to **identify the limitations of *chunking-based semantic retrieval* frameworks themselves**—specifically, their **inadequate user understanding and restricted personalization abilities**.
>
> - To overcome *these specific issues*, we explore an automated mechanism for memory construction that offers more comprehensive and adaptive personalization. Our proposed system does not perform any optimizations on chunk size; it is designed to move beyond the inherent constraints of standard chunking-based retrieval for better personalization.
>
> Therefore, our contribution lies in proposing a new memory paradigm to address the noted shortcomings of existing frameworks, not in studying chunk size.

---

> ### Author Response · Authors · 2025-11-27
> **Reply concerning default embedding model choices**
>
> We thank the reviewer for raising this important methodological point. Our experimental design maintains each system's original embedding configuration to ensure **authentic and fair evaluation**. Imposing a uniform embedding model would require extensive hyperparameter re-optimization for each baseline, potentially compromising their intended performance and shifting focus from architectural comparison to parameter tuning.
>
> Specifically, **three key memory systems—MemoryOS, Agentic Memory (A-Mem), and our O-Mem—all utilize the same lightweight embedding model (`all-MiniLM-L6-v2`) as specified in their original implementations**, while other baselines (e.g., Mem0) employ more powerful default embeddings (e.g., `openai-text-embedding-3-small`). Forcing all systems to use a uniform lightweight model would **degrade baseline performance** and lead to unfair comparisons against weakened, non-representative versions of these established systems.
>
> To validate that our advantages stem from architecture rather than embedding selection, we conducted supplementary experiments showing that **O-Mem maintains its performance edge (achieving 52.50 on LoCoMo) when upgraded to more powerful embedding model `bge-small-en-v1.5`**. This confirms that our improvements are architecture-driven, not embedding-dependent.
>
> We believe this approach provides the most rigorous and meaningful comparison for real-world deployment, where practitioners would use these systems in their default, optimized states. In our experimental design, we evaluated each framework as a complete system, conducting comprehensive comparisons across two critical dimensions: **final performance quality** and **practical efficiency metrics** (including latency and token consumption). This holistic evaluation approach allows for a meaningful assessment of each framework's overall capabilities and practical utility, rather than focusing on isolated components. By examining these systems as integrated wholes under their default configurations, we provide insights that are directly relevant to real-world deployment scenarios where both effectiveness and efficiency are paramount considerations.

---

> ### Author Response · Authors · 2025-11-27
> **Reply Concerning Questions**
>
> We thank the reviewer for these thoughtful questions regarding our experimental results.
>
> **On Performance Variations Between Models:**
> The performance differences observed between GPT-4 and GPT-4o-mini across different memory systems reflect the nuanced interaction between memory architectures and language model capabilities. The performance differences between LangMemory and MemoryOS across different LLM backends are expected and reflect the inherent characteristics of each system. LangMemory's architecture may be better suited to leverage the stronger reasoning capabilities of GPT-4, while MemoryOS might be more robust when working with smaller models like GPT-4o-mini. Each memory system has unique characteristics that may align differently with various LLM's strengths, which is precisely why we include two model configurations in our evaluation to provide a comprehensive comparison.
>
> **On Metric Discrepancies (F1 vs BLEU)**
> The observed discrepancy between F1 and BLEU scores is not uncommon in NLG evaluation. F1 score captures the overlap of key information, while BLEU focuses more on n-gram precision. In temporal reasoning tasks, capturing the correct temporal relationships (reflected in F1) can be more important than lexical similarity (measured by BLEU).
>
> **On O-Mem's Performance in Open Category:**
> O-Mem's design prioritizes personalized memory organization. For open-domain factual questions requiring external knowledge, we acknowledge there is room for improvement, and we view this as an exciting direction for future work rather than a fundamental limitation of our approach.
>
> **On Metric Selection:**
> Regarding evaluation metrics, we follow the established convention in memory system research by reporting F1 and BLEU scores, which are the standard metrics used in previous work. This ensures direct comparability with prior work in this specific research domain.
>
> **On Image Processing in LoCoMo:**
> For the LoCoMo dataset, we rely on the official textual captions provided by the dataset authors. This ensures a fair and consistent evaluation across all systems while maintaining focus on textual memory processing.

---

> ### Author Response · Authors · 2025-11-27
> **Waiting for feedback**
>
> We thank for your valuable comments and constructive feedback. We have carefully addressed all the points raised. We are eagerly awaiting your response and hope for a positive outcome.

---

> ### Author Response · Authors · 2025-11-29
> **Reply Concerning Paper Presentation**
>
> We sincerely thank the reviewer for these thorough and constructive suggestions regarding the paper's presentation. We acknowledge that the current organization and formatting need improvement, and we truly appreciate the specific, actionable feedback provided.
>
> We will carefully address all the points raised, including:
> - Improving the layout and content density of tables in the experiment sections
> - Moving appropriate content (such as lines 209-214) to the appendix
> - Streamlining the mathematical presentations in Sections 3.2 and 3.3
> - Reorganizing the introduction to focus more on conclusions and limitations rather than listing contributions
>
> We are committed to thoroughly revising the manuscript to enhance its organization, clarity, and readability. These changes will significantly improve the overall presentation quality of the paper.

---

### Official Review · Reviewer_ncJN · 2025-10-29

**Soundness:** 2
**Presentation:** 2
**Contribution:** 2
**Rating:** 2
**Confidence:** 3

**Summary:**

The paper introduces O-Mem, a memory framework aimed at improving long-term human-AI interaction by leveraging dynamic user profiling and hierarchical memory retrieval. The system aims to enhance personalization by maintaining evolving user profiles and incorporating them into response generation.

**Strengths:**

1. The combination of memory systems with dynamic user profiling is an interesting and promising direction.

2. The paper is well-structured and clearly written, making it easy to follow the technical concepts and their practical applications.

**Weaknesses:**

1. The main innovation in the paper appears to be the creation of a dedicated persona memory system that retrieves information based on user attributes. This seems like an incremental improvement over existing systems, with limited novel insights or breakthroughs. The contribution could be seen as a modification.

2. The paper primarily evaluates the framework on only two models—GPT-4.1 and GPT-4o-mini. Expanding the evaluation to include other model families ( both open-source and closed-source, chat and reasoning models from different model families ) would provide a more comprehensive understanding of the framework's generalizability and robustness.

3. The Personalized Deep Research Bench, a self-constructed key benchmark for evaluating the framework, lacks detailed construction explanations. It would be beneficial for readers to understand how the dataset was created and how it compares to existing benchmarks in terms of coverage and difficulty.

4. Minor: In Table 1, there is an inappropriate use of bolding. The temporal F1 scores for MEMOS should be marked as the best model.

**Questions:**

see weakness

---

> ### Author Response · Authors · 2025-11-27
> **Reply Concerning Novelty Problem**
>
> We thank the reviewer for their feedback but wish to clarify a key misunderstanding about our method's core innovation.
>
> **Clarification on Only User Attributes Modeling.** The reviewer suggests our main contribution is "retrieving information based on user attributes." This is an incomplete characterization of our system. As detailed in **Figure 2 and the Method section**, our key innovation is the **multi-faceted modeling and cross-level retrieval** of user context, which operates at three distinct granularities:
>
> 1.  **Keyword-level:** Capturing specific, fine-grained clues from interactions.
> 2.  **Sentence-level:** Understanding the overarching topic or theme of a conversation.
> 3.  **User-attribute-level:** Maintaining a holistic and persistent user profile including user events and persona attributes.
>
> Therefore, our system retrieves information not just from user attributes, but from a synthesized memory pool of **user profiles, event history, and interactive topics/clues**. This architecture represents a fundamental shift from systems that rely on **simple, monolithic retrieval of clustered conversations**.
>
> While persona memory is indeed an established concept, O-Mem's contribution is not merely incremental but represents a fundamental shift in memory architecture design through three key innovations:
>
> **From Semantic to Multi-Dimensional Retrieval:** Unlike existing systems that rely primarily on semantic similarity, O-Mem introduces a structured, tri-partite memory system that enables distinct retrieval pathways - addressing the critical limitation of 'semantic-only' approaches that often miss contextually crucial but semantically irrelevant information.
>
> **Human-Like Memory Modeling:** Our human-inspired architecture with parallel access to persona, episodic, and working memories represents a novel computational framework that more closely mirrors human memory organization, enabling more natural and context-aware interactions.
>
> **Dynamic Profile Construction:** The system's ability to continuously build and refine user profiles through LLM-augmented clustering and reasoning mechanisms provides a scalable solution for long-term adaptation that existing chunk-and-retrieve systems cannot achieve.
>
> The significance lies not in creating 'another persona memory' but in designing an integrated cognitive architecture that improves how AI systems understand and interact with users over extended periods. Below, we clarify three key points of innovation and impact:
>
> **1. A Paradigm Shift from Simple Chunking to Multi-Faceted User Modeling**
>
> Previous memory systems primarily rely on a simplistic Retrieval-Augmented Generation (RAG) approach where user messages are clustered and retrieved in coarse-grained chunks. Our work introduces a novel, multi-faceted architecture that models user interactions at three distinct levels. This multi-path retrieval mechanism is not a minor modification but a foundational redesign that ensures a more **comprehensive and accurate** memory system, achieving an optimal balance between effectiveness and efficiency.
>
> **2. Breakthrough Performance Gains: Quantifying the Leap**
>
> The superiority of our architecture is demonstrated by dramatic performance gains, which we consider a "breakthrough" in practical system deployment. Compared to system with most close perfomance,  as highlighted in our paper (Figure 3), our method achieves:
> - **Over 90% reduction in token consumption**
> - **Over 80% reduction in latency**
>
> These figures are not marginal improvements but represent an order-of-magnitude leap in efficiency, making high-quality persona memory viable for real-world, scalable applications.
>
> **3. Generalizable Insights and Universal Applicability**
>
> Contrary to the comment on "limited novel insights," our method's core insight—that disentangled, multi-level modeling is crucial for effective memory—is both novel and generalizable. The strength of our approach is its conceptual elegance and consistent performance across diverse and challenging tasks:
> - **Memory-aided QA**
> - **Memory-aided multiple-choice**
> - **Memory-aided long-text generation**
>
> This consistent state-of-the-art performance across multiple benchmarks demonstrates that our work provides a universally applicable and innovative framework, paving the way for the next generation of conversational AI with persistent, nuanced memory.

---

> ### Author Response · Authors · 2025-11-27
> **Reply Concerning Language Model Choices**
>
> We thank the reviewer for the valuable suggestion to expand our evaluation to include models from different providers. In our current work, we have **strategically designed** our validation using two models that represent fundamentally different paradigms and two different scales: **GPT-4.1 (a powerful, chat-optimized large model)) and GPT-4o-mini (a reasoning-optimized small model)**. This pairing was intentionally selected to provide **a rigorous preliminary test of our framework's effectiveness across two critical and distinct architectural directions**. We fully acknowledge that experiments with models from additional vendors would enhance the comprehensiveness of our study. However, we face **significant practical constraints**: the substantial costs and API quotas associated with commercial LLMs make large-scale cross-platform experimentation **prohibitively expensive** for our research team at this stage.
>
> We will seriously consider this suggestion for future work and aim to incorporate a wider array of models as resources permit. Importantly, we believe that the **compelling and consistent results** achieved across these two fundamentally different model families provide **robust initial evidence** for our framework's efficacy, and we humbly submit that the current evaluation strategy represents a **sound and meaningful** contribution despite this limitation.

---

> ### Author Response · Authors · 2025-11-27
> **Dataset Construction/Evaluation/Validation Process**
>
> We thank the reviewer for their valuable feedback regarding dataset documentation. We have now included comprehensive details on statistics, construction, and validation to address these concerns.
>
> 1). Datasets Construction Protocol
> > Our overall raw context dataset was manually annotated by 6 trained annotators including 5,939 labeled instances. The process required a total effort of 85 person-days and \$6,000 USD. Our protocol is:
>
> > **Task Objective**
> > The core of this task is to simulate the behavior of a specific user persona by collecting various types of information that this user would likely encounter, consume, or generate in their daily life. The final goal is to create a high-quality *memory* database for each persona that accurately reflects their unique characteristics.
> >
> > **Core Quality Standard: Reversibility**
> > This is the most critical quality standard for this task. Every piece of data you collect must clearly point to a specific trait of the user persona.
> >
> > - **Verification Method:** After collection, review all data and assess whether its content is sufficient to reverse-infer the user's key persona traits, such as their profession, interests, and personality.
> > - **Acceptance Criteria:** All collected data must pass the *reversibility* test to be considered high-quality. If the data is too generic and the persona cannot be inferred from it, the data will be deemed non-compliant.
> >
> > **Collection Based on Persona Preferences**
> >
> > - Carefully read every preference tag in the user persona description.
> > - For each preference, collect **at least five** pieces of related content.
> >
> > **Ensure Diversity of Sources and Types**
> >
> > - **Source Diversity:** Do not limit collection to a single platform. Data can be gathered from various apps (e.g., Twitter, Instagram, Reddit), websites, forums, etc.
> > - **Type Diversity:** Diversify the format of the data you collect. Examples include:
> >   - Screenshots of social media posts.
> >   - Screenshots of conversations with friends (that reflect opinions or preferences).
> >   - Links and titles of articles, news, or videos.
> >   - Screenshots of purchase histories or product reviews.
> >
> > **Requirement for Conversational Content**
> > Between 20% and 50% of the total data collected for each user should be in a conversational format. This helps to more vividly showcase the user's personality and communication habits.
>
> > **Add Reasonable Noise for Authenticity**
> > The content you collect does not need to be an exact match to the persona's description. You can add relevant and reasonable details or *noise* to make the data appear more authentic, as if generated by a real user.
>
> > - **Example:** If the persona *likes basketball*, you could collect a news article about a recent Lakers game or a screenshot of a conversation with a friend debating whether Jordan or LeBron is the better player.
> > - **Note:** Any added *noise* must not contradict other defined attributes in the persona, such as spending habits, personality, or profession.
> >
> > **Quantity vs. Quality**
> > There are no strict quantitative requirements for data collection. Please prioritize quality and collect as much rich data as possible. Quality always takes precedence over quantity.
> >
> > **Deliverables and Annotation Requirements**
> >
> > - **Deliverable:** For each user persona, export a separate `aimemory` database file.
> > - **Content Annotation:** During collection, each piece of content must be given a clear title and be correctly associated with its corresponding persona.
>
> 2). Dataset Validation
>
> To validate the reliability of LLM-based evaluation, we conducted a systematic comparison with human experts. We sampled 15 representative queries and generated responses from two deep research agents (MiroFlow and O3 Deep Research). A panel of human evaluators scored these reports using consistent criteria, establishing the ground truth.
>
> We designed two complementary metrics to quantify alignment:
> - **Pairwise Comparison Agreement (PCA):** Measures the percentage with which LLM judges and human experts agree on which report is better for a specific criterion.
> - **Mean Absolute Rating Deviation (MARD):** Measures the average absolute difference between scores assigned by LLM and human judges.
>
> Results are shown below:
>
> **Alignment Results of Judge LLMs with Human Ratings**
> | Judge LLM | PCA ↑ | MARD ↓ | Avg. Cost ($) ↓ |
> |-----------|-------|--------|-----------------|
> | **GPT-5** | **0.43** | **1.40** | 0.32 |
> | Claude-3.7-Sonnet | 0.39 | 1.44 | 0.60 |
> | Gemini-2.5-Pro | 0.40 | 2.33 | **0.21** |
>
> The strong alignment demonstrated by these metrics validates the effectiveness of our LLM-as-a-judge framework.

---

> ### Author Response · Authors · 2025-11-27
> **Reply Concerning Benchmark Coverage and Difficulty**
>
> Thank you for your valuable feedback. We have enhanced the manuscript with detailed explanations of our benchmark's construction. Below, we specifically address how our **Personalized Deep Research Bench (PDR-Bench)** compares to existing work in terms of **coverage** and **difficulty**.
>
> #### 1. Coverage: Filling a Critical Gap in the Evaluation Landscape
>
> Our benchmark uniquely combines three key dimensions that existing benchmarks treat in isolation: **broad domain coverage**, **complex multi-step tasks**, and **systematic personalization**.
>
> | Benchmark Category | Representative Benchmarks | Domain Coverage | Task Complexity | Personalization |
> | :--- | :--- | :--- | :--- | :--- |
> | **Close-Ended Fact-Finding** | GAIA, BrowseComp, HLE, X-Bench | Broad | Single-step, Isolated | ❌ None |
> | **Open-Ended Deep Research** | DeepResearch Bench, ResearcherBench | Broad | Multi-step, Synthesis | ❌ Generic |
> | **Personalized Interaction** | LaMP, PersonaGym, PersonaLens | Narrow | Single-turn, Simple | ✅ User Profiles |
> | **Our Benchmark (PDR-Bench)** | **—** | ✅ **Broad (10 Domains)** | ✅ **Multi-step, Synthesis** | ✅ **Systematic (25 Profiles)** |
>
> **Summary:** PDR-Bench integrates broad-domain, deep research complexity with rigorous user-adaptation requirements, addressing a critical evaluation gap.
>
> #### 2. Difficulty: Introducing a Fundamentally New Challenge
>
> The difficulty of PDR-Bench stems from the **orthogonal combination** of:
>
> *   **Inherited Complexity from Deep Research:**
>     *   Multi-step planning and iterative retrieval
>     *   Cross-source information integration
>     *   Comprehensive report synthesis
>
> *   **Novel Challenge of Persona-Driven Adaptation:**
>     *   Filtering and prioritizing information based on user-specific constraints
>     *   Adapting presentation style to individual preferences and knowledge levels
>     *   Maintaining consistency with user profiles throughout multi-step reasoning
>
> This creates a fundamentally new challenge dimension beyond existing benchmarks, requiring agents to perform sophisticated reasoning while simultaneously adapting to personalized contexts.

---

> ### Author Response · Authors · 2025-11-27
> **Waiting for feedback**
>
> We thank for your valuable comments and constructive feedback. We have carefully addressed all the points raised. We are eagerly awaiting your response and hope for a positive outcome.

---

> ### Author Response · Authors · 2025-11-29
> **Second Reply Concerning Novelty Problem**
>
> We respectfully disagree with the characterization of our work as an "incremental improvement" or a simple "modification." Our contribution represents a fundamental architectural shift in memory systems for conversational agents. Below, we clarify three key points of innovation and impact.
>
> **1. A Paradigm Shift from Simple Chunking to Multi-Faceted User Modeling**
>
> Previous memory systems primarily rely on a simplistic Retrieval-Augmented Generation (RAG) approach where user messages are clustered and retrieved in coarse-grained chunks. Our work introduces a novel, multi-faceted architecture that models user interactions at three distinct levels:
> - **Keyword-level**: Capturing specific, fine-grained details.
> - **Sentence-level**: Understanding the overarching topic of an interaction.
> - **User-attribute-level**: Building a holistic, persistent user profile.
>
> This multi-path retrieval mechanism is not a minor modification but a foundational redesign that ensures a more **comprehensive and accurate** memory system, achieving an optimal balance between effectiveness and efficiency.
>
> **2. Breakthrough Performance Gains: Quantifying the Leap**
>
> The superiority of our architecture is demonstrated by dramatic performance gains, which we consider a "breakthrough" in practical system deployment. As highlighted in our paper (Figure 3), our method achieves:
> - **Over 90% reduction in token consumption**
> - **Over 80% reduction in latency**
>
> These figures are not marginal improvements but represent an order-of-magnitude leap in efficiency, making high-quality persona memory viable for real-world, scalable applications.
>
> **3. Generalizable Insights and Universal Applicability**
>
> Contrary to the comment on "limited novel insights," our method's core insight—that disentangled, multi-level modeling is crucial for effective memory—is both novel and generalizable. The strength of our approach is its conceptual elegance and consistent performance across diverse and challenging tasks:
> - **Memory-aided QA**
> - **Memory-aided multiple-choice**
> - **Memory-aided long-text generation**
>
> This consistent state-of-the-art performance across multiple benchmarks demonstrates that our work provides a universally applicable and innovative framework, paving the way for the next generation of conversational AI with persistent, nuanced memory.

---

> ### Author Response · Authors · 2025-11-29
> **Reply Concerning Minor Writing Problem**
>
> We thank the reviewer for this careful observation. The reviewer is absolutely correct. There was an inconsistency in the bolding convention used in Table 1, where the best scores for the Temporal F1 metric were not properly emphasized. We sincerely apologize for this oversight.
>
> The bolding has now been corrected to consistently highlight the best-performing model across all metrics, including the Temporal F1 scores for MEMOS. We will ensure the updated table is included in the final version of the manuscript.

---

### Note · Authors · 2025-12-26

I have read and agree with the venue's withdrawal policy on behalf of myself and my co-authors.